# DFTrans: Dual Frequency Temporal Attention Mechanism-Based Transportation Mode Detection

**DOI:** 10.3390/s22218499

**Published:** 2022-11-04

**Authors:** Pu Wang, Yongguo Jiang

**Affiliations:** College of Computer Science and Technology, Ocean University of China, Qingdao 266404, China

**Keywords:** deep learning, pattern recognition, activity recognition, transportation mode detection

## Abstract

In recent years, with the diversification of people’s modes of transportation, a large amount of traffic data is generated when people travel every day, and this data can help transportation mode detection to be of great use in a variety of applications. Although transportation mode detection has been investigated, there are still challenges in terms of accuracy and robustness. This paper presents a novel transportation mode detection algorithm, DFTrans, which is based on Temporal Block and Attention Block. Low- and high-frequency components of traffic sequences are obtained using discrete wavelet transforms. A two-channel encoder is carefully designed to accurately capture the temporal and spatial correlation between low- and high-frequency components in both long- and short-term patterns. With the Temporal Block, the inductive bias of the CNN is introduced at high frequencies to improve generalization performance. At the same time, the network is generated with the same length as the input, ensuring a long effective history. Low frequencies are passed through Attention Block, which has fewer parameters to capture the global focus and solves the problem that RNNs cannot be computed in parallel. After fusing the output of the feature by Temporal Block and Attention Block, the classification results are output by MLP. Extensive experimental results show that the DFTrans algorithm achieves macro F1 scores of 86.34% on the real-world SHL dataset and 87.64% on the HTC dataset. Our model can better identify eight modes of transportation, including stationary, walking, running, cycling, bus, car, underground, and train, and has better performance in transportation mode detection than other baseline algorithms.

## 1. Introduction

In recent years, there have been many new applications of transportation mode detection in various aspects of daily human life. Examples include the detection of urban transportation mode detection [1], the development of context-aware applications [2], carbon footprint calculations [3], and the mining of population movement patterns [4]. Humans undertake multiple modes of transportation daily and the wealth of sensor data that many models of mobile phones can collect means that there is a wealth of traffic-related data in the real world.

IMU data can be passively accessed through operating systems or applications. Transportation mode detection can be applied to reduce CO_2_ consumption [3], mine population movement patterns, predict future traffic conditions, etc. Some research centers have organized experiments with volunteers to study the mining of population mobility patterns by recording all their traffic patterns in data form through smartphone sensors on their smartphones [4]. Researchers can build traffic systems that, given an arbitrary time and place, can predict future traffic conditions [5]. In addition, transportation mode detection can provide contextual clues to the travel behavior of a city by integrating the geography in which mobility occurs through high-resolution (spatial and temporal) mobility data [6].

In this paper, we propose a more accurate and efficient data training model, DFTrans, based on the ideas of Temporal Block for transportation mode detection and the Attention Block model that has worked very well recently. We test and evaluate the performance of the DFTrans model on the SHL dataset [7] and the HTC dataset [8] in a variety of situations. These datasets contain three inertial sensors and one barometric sensor, each with three elements in the X, Y, and Z axis. The data is collected via four sensors: the accelerometer, gyroscope, magnetometer, and barometer. A detailed illustration of the sensor data is shown in Section 4.2. It is worth pointing out that our study removes the limitations of previous motion sensors or GPS [4,6,9] and uses IMU data to escape the limitations of GPS in closed spaces with poor signals and a single method of data collection. Many previous studies of transportation mode detection have simplified transportation modes into four categories: stationary, walking, cycling, and by car [10]. To better exploit the value of transportation mode detection in practical applications, and in contrast to previous studies, we distinguish this time between a more subdivided set of eight modes of transport: stationary, walking, running, car, bus, cycling, underground, and train. Although many previous studies have used traditional machine learning [11,12,13,14], our DFTrans model is a deep-learning algorithm for automatic feature extraction.

The relevance and validity of the dataset greatly influence the upper limit of the value of transportation mode detection in practical applications. Roy et al. [6] used GPS data, incorporating the geographic context in which mobility occurs, to assess the role of geography in transportation mode detection. Wang et al. [15] selected seven GPS-related variables from an initial list of 22 variables as a feature set to develop and evaluate a random forest classifier combined with a rule-based approach to transportation mode detection. Bryan et al. [16] combined downscaling and classification algorithms with data collected from GPS and accelerometer sensors for transportation mode detection that could be used on a typical smartphone.

However, the high power consumption of GPS, the signal instability of GPS, and the singularity of the collected data have limited the development of transportation mode detection-related applications. More and more researchers are adopting inertial sensors on smartphones and introducing deep-learning techniques for transportation mode detection. H. Liu et al. [17] proposed an end-to-end bi-directional LSTM classifier-based trajectory transport. Li [18] et al. proposed an independent recurrent neural network (IndRNN) to process data of different lengths to better address the distinction between “trains” and “buses”. Sharma A et al. [19] used a classifier-based deep-learning model to define a de-decision strategy for the prediction of transport patterns for incoming time series, none of which worked well. The optimization of transportation mode detection models still faces significant challenges.

Many of the current technologies are still based on GPS and WiFi, but the common problems are excessive energy consumption and poor stability in enclosed spaces. Secondly, the uncertainty of road conditions and differences in driving habits during motor vehicle travel can also make it difficult to distinguish between the two modes of transportation, trains, and subways.Due to the lack of feature extraction from CNNs, Attention Block performs poorly in generalization without pre-training or with small datasets.Most current transportation mode detection efforts use a single approach, utilizing complex temporal patterns (e.g., short-term thunderstorms and long-term daily trends) that do not accurately capture the spatial and temporal dependencies of the different patterns.

Much of the current work makes use of temporal patterns of entanglement within a single model and fails to accurately extract spatiotemporal information across different patterns. Recently, a Transformer-based model [20] has been introduced for vision-related tasks. Based on its great success in vision tasks, we used Attention Block to process continuous data and capture global dependencies. In our study, based on Temporal Block and Attention Block, we design an algorithm DFTrans to study the transportation mode detection problem. The contributions of this paper are summarized as follows:We propose the DFTrans model, which is a deep-learning model based on Temporal Block and Attention Block. In the temporal dimension, long-term and short-term temporal patterns are separated using discrete wavelet transforms to avoid interference between them. We use Temporal Block and Attention Block to capture the short-term and long-term temporal correlations in the high- and low-frequency components, respectively. The fluctuating high frequencies are passed through the Attention Block to fully capture the high-frequency features, and the stable, low-frequency components with long-term trends are fed into the Temporal Block to fully capture the high-frequency features.We can distinguish well between eight specific modes of transportation: stationary, walking, running, cycling, car, bus, subway, and train. DFTrans is used for transportation mode detection with multiple lightweight sensors integrated into mobile phones. It improves the accuracy of transportation mode detection and increases the scalability of DFTrans for different combinations of multiple sensors.Parallel implementation of convolution via Temporal Block has a long and effective history while alleviating the Attention Block’s lengthy attention backbone design, which leads to feature richness and limited training samples for a fixed computational budget. It also compensates for the Attention Block’s lack of CNN functionality while remaining lightweight.To evaluate the performance of DFTrans, we set up baselines including decision trees, random forest, XGBOOST, convolutional neural network, multilayer perceptron, LR + MLP, Bi-LSTM, CNN + LSTM, temporal convolutional network, and Vision Transformer. DFTrans achieved an average accuracy of 86.69% on the SHL dataset and 87.51% on the HTC dataset. Experimental results showed that DFTrans outperformed the performance of the baseline algorithm, on the SHL dataset and HTC dataset, respectively, over the best baseline by improved by 1.73% and 2.08%.

The rest of our paper is organized as follows: We discuss some related work on transportation mode detection in Section 2, including some deep-learning methods based on GPS data or inertial sensor data. Section 3 describes the architecture of the DFTrans algorithm and then details the design details of DFTrans. Section 4 describes in detail the two datasets and the baselines that we used. We then present and discuss the evaluation results of DFTrans through experiments. Finally, Section 5 summarizes our work and describes plans.

## 2. Related Work

As GPS positioning is well-established and widely used on mobile phones, many researchers have worked on transportation mode detection using GPS. S Dabiri [21] et al. used a convolutional neural network architecture to predict traffic patterns based on GPS data and achieved the highest accuracy of 84.8%. A Roy [6] developed a data-driven modeling framework to detect traffic patterns using GPS data and secondly, to evaluate how the accuracy and generalization ability of the model changes with the addition of geographic context. J Li et al. [22] used the GeoLife GPS trajectory dataset. They proposed a feature extraction and machine learning-based classification algorithm to distinguish between walking, cycling, bus, car, and underground. Based on the GPS sensors on the user’s mobile device and the underlying traffic network knowledge, L. Stenneth et al. [23] tested five different algorithmic models, including Bayesian networks, DT, RF, MLP, and plain Bayes, in their experiments for transportation mode detection. Although GPS data is used by several research institutes [9], some of the limitations of GPS itself have affected many research advances in the field.

GPS has the problem of high energy consumption in addition to shadowing poor signals in confined spaces. As a result, some researchers have started to use datasets collected in the form of sensors. I. Drosouli et al. [2] used smartphone-embedded sensors (accelerometers, magnetometers, etc.) using a DR feature extraction algorithm to achieve excellent classification results. Elena et al. [24] used data from smartphone sensors, using different supervised learning methods as building blocks, to develop a hierarchical classification framework for transportation mode detection that outperforms traditional classifiers. In most machine learning, manual extraction of features is a significant burden, and accuracy and robustness could be improved. Some other physical layer security methods, like RF fingerprinting techniques, have the potential to be introduced for transportation systems [25,26].

Research on deep-learning models has been rapidly developed and applied in recent years. The deeper the non-linear modeling capability of deep learning compared to traditional machine learning, the better the performance for realistic and complex tasks. You. Majeed et al. [27] performed a vanilla segmentation learning model and demonstrated that the segmentation neural network (SplitNNN) performs similarly to the baseline. C Wang [28] et al. proposed an algorithm based on residuals and LSTM recurrent networks using inertial sensor data from mobile phones combined with an attention model to improve the accuracy of transportation mode detection. L. Wang et al. [29] were to identify eight traffic modes from smartphone inertial sensors, demonstrating applying various classical methods. H Liu [17] et al. proposed an end-to-end bi-directional LSTM classifier-based trajectory transportation mode detection framework using data collected by GPS, which automatically learns features from trajectories and achieves high accuracy in transportation mode detection. C Ito et al. [30] used 5-s FFT spectrogram images of accelerometer and gyro sensor data as training data. They applied migration learning methods to increase the accuracy of transportation mode detection. Y Qin et al. [31] used convolutional neural networks to learn feature representations. Then the LSTM network further learned the time-dependent features of the CNN output feature vectors to accurately identify walking, running, cycling, driving, and taking the bus, underground, and train. Samuel Ricord [32] et al. used Bluetooth and WiFi sensing data to calculate travel times for different modes of travel and proposed a linear model for new travel time calculation methods for pedestrians, bicycles, and cars by weighing travel times according to the highest, lowest and most probable speeds, with a transportation mode detection. An accuracy of about 83% was achieved. Z Chen [33] et al. proposed a bidirectional long- and short-term memory (ABLSTM) that uses WiFi CSI signal data to distinguish between six human activities. J Ding [34] et al. proposed a deep recurrent neural network approach to recognize transportation mode detection based on WiFi CSI data. The use of WiFi signals for transportation mode detection is relatively new but requires more stringent density balancing of WiFi access points and a more homogeneous form of data. Although there are some results from these efforts in transportation mode detection, there are also some shortcomings, which we have collated into a table, as shown in Table 1:

In contrast to previous studies, we propose a DFTrans model based on Temporal Block and Attention Block, which performs transportation mode detection through smartphone sensor data. DFTrans accurately captures the temporal and spatial correlation between low- and high-frequency components of long- and short-term patterns. In contrast to other studies, an inductive bias of the CNN is introduced at high frequencies to improve generalization performance. At the same time, the network is generated at the same length as the input to ensure a longer effective history. Low frequencies are passed through the Attention Block, which has fewer parameters to capture the global focus and also addresses the problem that RNNs cannot be computed in parallel. Extensive experimental results show that DFTrans achieves macro F1 scores of 86.34% on the SHL dataset and 87.64% on the HTC dataset.

## 3. Algorithm

### 3.1. Overview of the DFTrans Model

Our proposed DFTrans model, shown in Figure 1, consists of a Multimodal Input Layer, a Disentangling Flow Layer, a Temporal Attention Layer, and an MLP Layer. We use the barometric pressure sensor on the smartphone and inertial sensors: linear acceleration, gyroscope, and magnetometer, each containing three elements, including the x, y, and z axes. In the data pre-processing step, all datasets are transformed into a uniform matrix and fed into the Multimodal Input Layer to optimize the trainable parameters in the DFTrans model. The pseudocode of the detailed algorithm is shown in Algorithm 1. The function of Disentangling Flow Layer is to avoid interference between the low- and high-frequency components by separating them from the entangled time series in long- and short-time modes.

On the other hand, the high frequencies of the fluctuations have short-term effects and are fed into the Temporal Block for the convolution operation of the temporal convolution network. Multi-scale feature extraction facilitates the causality of temporal features and broadens the perceptual field. The stable low frequencies obtained by DWT processing have a long-term trend. Attention Block exploits its ability to capture global background information with attention, building dependence on the target and extracting more powerful features. Finally, the two fused features are fed into an MLP Layer with time-dependent representations and produce transport mode estimates using Softmax for classification. DFTrans uses the test data to evaluate performance based on accuracy, precision, recall, and F1 scores.

### 3.2. DFTrans Model

In this section, we present the details of how each part of the DFTrans model is handled.
Multimodal Input Layer. The pre-processed four sensor data are defined as tensor $A_{n,\;d,k}$ and fed into DFTrans from the Multimodal Input Layer, where n denotes the total number of samples, k denotes the total number of cells for all sensors, and d denotes the length of the selected sliding window. In the SHL dataset used in this paper, d = 500 corresponds to a sampling period of 5 s and a sampling frequency of 100 Hz. The k = 10 represents the linear acceleration axis, gyroscope axis, magnetometer axis, X axis, Y axis, and Z axis of each of the three inertial and barometric sensors. The three elements of the three inertial sensors are fused and fed into the Disentangling Flow Layer. In contrast, the barometric sensor tensor is fed directly into the Disentangling Flow Layer, as shown in Figure 2.
**Algorithm 1** DFTrans model
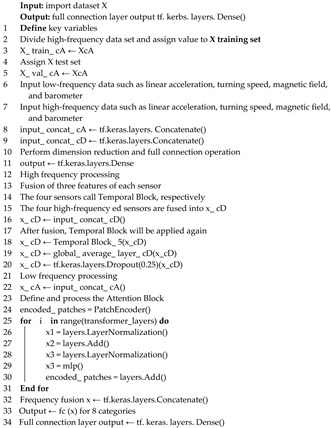
Disentangling Flow Layer. After the four sensor tensors are fed into the Disentangling Flow Layer, the discrete wavelet transform (DWT) is performed separately to obtain the low-frequency and high-frequency components from the entangled traffic sequence. The low and high frequencies from each of the four sensors are fused and fed into the temporal attention layer. The low-frequency components of the sensor data have a stable and long-term trend, whereas the fluctuating high-frequency components of the sensor data are full of short-term effects. The discrete wavelet transform of the input sequence X can be expressed as:(1)Xl=gX,Xh=hX
where g is a low-pass filter and h is a high-pass filter. The scaling function and wavelet function of the discrete wavelet are closely related to the low-pass filter g={gk}k∈ℤ and high-pass filter h={hk}k∈ℤ, respectively. The time steps of the low- and high-frequency components Xl and Xh are reduced to half the input by the downsampling operation of the DWT.Temporal Attention Layer. The Temporal Attention Layer mainly consists of Temporal Block and Attention Block, low-frequency components are sent to Temporal Block, and high-frequency components are sent to the Attention Block. Temporal Block is a convolutional neural network model variant that combines causal convolution, extended convolution, and residual linking on top of a 1D convolutional neural network. Temporal Block: With a flexible perceptual field and a stable gradient, it is possible to map temporal data to an output sequence of the same length [35]. Temporal Block uses a 1D convolutional kernel to scan the current time node and previous historical time series. The input sequence is x0,…, xT, and at each period, the output y0,…, yT. A sequence modeling network is any function f:XT+1→YT+1 that produces the mapping:(2)y^0,…,y^T=f(x0,…,xT)


Like RNNs, Temporal Blocks have a flexible perceptual domain and a stable gradient that allows the output sequence length to remain constant, exhibiting longer adequate memory. We now describe in detail the structure of the Temporal Block:Temporal Block uses causal convolution. The output at this moment is only convolved with the input from the previous moment, independent of the future moment. The Temporal Block adds an extra zero-padding length to keep the subsequent layer the same length as the previous layer, as shown in Figure 3.Temporal Block uses extended convolution. The receiving domain of the data is expanded by setting the size of the kernel and setting the values of the expansion coefficients for the different layers, as in Figure 3a. More formally [31], for a one-dimensional sequence of inputs x∈ ℝ and a filter f:{0,…,k−1}→ℝ, the extended convolution operation F on the sequence element s is defined as:(3)$$F\left(s\right)=\left(x\times df\right)\left(s\right)=\sum_{i=0}^{k-1}f\left(i\right)\cdot x_{s-d\cdot i}$$
where  d is the expansion coefficient, k is the filter size, and s−d·i denotes the past direction.Temporal Block introduces a residual network. A branch of the residual block produces output through transformations F, which is then added to the input x. The residual block alleviates the problem of gradient disappearance in deeper models by “connecting” one or more separated network layers [36]:(4)o=Activation(x+ℱ(x))

We utilize a residual module instead of a convolutional layer, as shown in Figure 3b. In addition, an extra 1 × 1 convolution is added to the Temporal Block, keeping the elements summed to give the same shape tensor.

The Attention Block receives the low-frequency components from the Disentangling Flow Layer. It uses the low-frequency components of the ten elements of all sensors as sequence input, denoted as $A_{n,\;a,1}$, where n denotes the total number of samples and b denotes the length of the selected sliding window, after DWT a=d/2. Like BERT’s [class] token, we use a one-dimensional positional embedding added to the patch embedding to preserve positional information. The resulting sequence of embedding vectors is used as input to the encoder, as shown in Figure 4.

In the Attention Block, only the MLP layer is local and trans-equivalent, whereas the self-attentive layer is global. The transformer encoding consists of alternating layers of multi-headed self-retention and MLP blocks, as shown in Figure 5. Separate Layer Norm and residual connections will be applied before and after each block. The Attention Block feature and output learn the low-frequency components from the transformer encoding.

4.MLP layer. The low-frequency component data features processed by the Temporal Attention Layer are fused with the high frequency component data features and fed into an MLP layer consisting of four fully connected networks. We use the dropout method [37] in the MLP layer to reduce the overfitting problem [38]. We use Equation (5) to define the output of the ith fully connected layer:


(5)
$$o_{i}=\;\mathrm{Activation}\;\left(W\cdot x+b\right)$$


We define *W* as the weight of the hidden layer and *b* as the deviation.

The cell numbers of all hidden layers are 64, 128, 256, and 8. The activation functions of the first three layers are all ReLU, and the activation function of the last layer is Softmax. The drop probabilities of the first three hidden layers are set to 0.5 and the L2 regularizer of hidden layer II is set to 0.02. Output $A_{n,\;8}$ from hidden layer V, as shown in Figure 6.

## 4. Experimental Evaluation

In this section, we detail the SHL dataset and HTC dataset that we used and the pre-processing process for both datasets. The more popular baselines algorithm is then experimented with on both datasets, and the detailed parameter settings for the baselines algorithm are listed in the text. Finally, the performance of the DFTrans and baselines algorithms are compared in multiple dimensions, and the experimental results are explained and evaluated in detail for reasons.

### 4.1. Introduction to the Dataset

SHL Dataset: The SHL dataset [7] is traffic data formed by three UK volunteers over a seven-month period in 2017. Eight modes of transport were tagged during each day of traffic. These samples were recorded using four Huawei Mate 9 smartphones, placed in a bag, on the wrist, strapped to the chest, and placed in a pocket. The four smartphones were equipped with an application for recording data. The dataset was organized by each user (User1,….User4) by record date (e.g., 150,321 on 21 March 2015) and stored in a hierarchical structure. All files associated with a date are stored in the corresponding directory. In particular, the SHL dataset contains three inertial sensors and one barometric sensor, each with three elements in the X, Y, and Z axis, so M = 10. In our study, only lightweight sensor data (i.e., accelerometer, gyroscope, magnetometer, and barometer) were used to evaluate the performance of DFTrans to identify the eight modes of traffic. We chose approximately 272 h of sensor data collected by the same volunteer to train and test our DFTrans model. Each sensor was sampled at a frequency of 100 Hz. To make better use of the time dependency of DFTrans, we reorganized the temporal order of the dataset.HTC Dataset: We used the HTC dataset collected by 150 volunteers using smartphones to further evaluate the scalability of the DFTrans model [8]. The HTC dataset contains 8311 h and 100 GB of data since 2012. The same kind of sensors was used for each sample pack and SHL. To keep the two datasets of the same classification kind, we filtered out the motorbike and SHL data. Different sensor data with timestamps of less than 0.1 s were defined.

### 4.2. Data Preprocessing

We use a sliding window to split the dataset into individual ‘sequences’ which are used as input to the DFTrans model. The raw data is divided into a series of fixed-length sequences, and the eight windows that are divided correspond to the eight traffic patterns to be distinguished in this paper. Figure 7 and Figure 8 show the *x*-axis data for linear acceleration in the SHL dataset and the HTC dataset, respectively.

To reduce the interference of various noises and errors in the raw sensor data, we performed operations to remove dirty data and normalize the data for different sensor data, which can have a more effective improvement of the experimental performance of DFTrans.

Dirty data removal: The dataset we used had some samples of sensor vector elements that were missing one or two elements. However, given the large dataset used in our study, we applied a low-cost deletion operation to the sensor data.Normalization: In this experiment, we performed a Z vector normalization operation [39] on each element of the vector data for all inertial sensors to reduce the significant differences in data ranges between heterogeneous sensors:(6)$$x^{\prime }=\frac{x-\mu }{\sigma }$$
where μ and σ are the mean and standard deviation of each element of the sensor data, respectively.

### 4.3. Baseline

In order to more objectively evaluate the multidimensional results of DFTrans, we used several popular algorithms as baselines, including machine learning algorithms (i.e., DT, RF, and XGBOOST), as well as more popular deep-learning algorithms (i.e., CNN, CNN + LSTM, Bi-LSTM, TCN, and Vision Transformer). In these baselines, some of the strengths of TCN and Vision Transformer were borrowed from our DFTrans model. The parameters for DT, RF, and XGBOOST are all defaults in the MATLAB Machine Learning Toolbox and the parameters that were changed are mentioned explicitly. Table 2 shows the parameters of baselines with special settings.

DT: Starting from the root node, instances are tested and allocated recursively. Finally, the instances are divided into the classes of the leaf nodes [39].RF: Random forest contains multiple decision tree classifiers which evaluate the importance of variables when deciding on a category [40].XGBOOST: XGBOOST is primarily designed to reduce bias, that is, to reduce the error of the model [41]. It continuously reduces practical differences by learning from multiple learners.CNN: The convolutional neural network consists of a convolutional layer, a pooling layer, and a fully connected layer, and can also be trained using a backpropagation algorithm.MLP: A multilayer perceptron (MLP) has multiple hidden layers and demonstrates a high degree of connectivity.LR + MLP [42]: Linear model LR and nonlinear MLP neural network models are applied to improve the predictive power of the models.Bi-LSTM [17]: A recursive neural network approach is used. A bidirectional LSTM architecture based on rotation and translation invariance for training is proposed.CNN + LSTM: The algorithm combines a CNN, which learns feature representations, and an LSTM unit, which is applied to the output of the convolutional neural network [43].TCN [44]: Algorithms used to solve time series predictions, like RNN. Predictions can be made using a combination of very deep networks and dilated convolutions.Vision Transformer [20]: Vision Transformer (ViT) model structure mainly consists of patch embedding, transformer encoder, and MLP head, which have good classification performance.

### 4.4. Metrics

The experimental results of DFTrans were evaluated using four metrics: accuracy, recall, precision and F1 scores, defined as Equation (7):(7)$$F1=2\times \frac{\;\mathrm{Recall}\;\times \;\mathrm{Precision}}{\;\mathrm{Recall}+\mathrm{Precision}}$$

### 4.5. Experimental Settings

We trained the DFTrans model with pre-processed SHL datasets and HTC datasets using the Keras deep-learning framework. We optimized using the Adam [45] optimizer and a cross-entropy loss function with a learning rate = 0.001. The model was trained with epoch parameter = 140 and batch size = 32. We trained DFTrans on a PC with GPU and Table 3 shows the details of the configuration of our experimental equipment.

### 4.6. Performance Evaluation of Baselines on Two Datasets in Experiments

As shown in Table 4 and Table 5, the results of all deep-learning baselines and DFTrans outperform the results of machine-learning models because TCN and DFTrans make use of the convolutional neural network in terms of feature extraction. The superiority of deep learning’s ability to automatically extract features is demonstrated on larger data sets. The average precisions for DFTrans were 0.4% and 2.25% higher than the best performing TCN in the baseline on the two datasets because, although TCN can identify very long time histories, TCN utilizes complex temporal patterns and does not accurately capture the spatial and temporal dependencies of different patterns. MLP and LR + MLP are more than 70% accurate, but it is difficult to distinguish effectively between subways and trains. Using these baselines, you can’t learn high-level functions and time dependencies. CNN + LSTM also performs well, and convolutional neural networks can learn appropriate feature representations, and feature expressions are robust for transportation mode detection. An LSTM unit is used at the output of the CNN and improves the performance of transportation mode detection.Our proposed DFTrans significantly outperformed the other baselines. The F1 scores of DFTrans on the two datasets were 86.34% and 87.64%, respectively. DT, RF, and XGBOOST could accurately identify six modes of transport, i.e., stationary, walking, running, cycling, bus, and car. However, the accuracy was lower for trains and subways. The accuracy of all baselines was above 70%. In contrast, DFTrans makes a good distinction between all eight modes of transport. The convolution of the Temporal Block in DFTrans is causal. It can map high-frequency components of sensor data to output sequences of the same length, allowing the prediction of very deep networks. The Attention Block is better at extracting global features and learning the features of the low-frequency components of the sensor data.Table 6 and Table 7 show the performance of recently popular machine learning and deep-learning algorithms in terms of accuracy on the two datasets, respectively. On the SHL dataset, the accuracy of DFTrans is 86.69%, which is 1.73% higher than the best-performing TCN algorithm [46]. On the HTC dataset, the average accuracy of DFTrans is 87.51%, which is 2.08% higher than the best-performing TCN in baselines. DL contains four classifiers, and whereas ML does not have as high an upper limit as DL, ML’s algorithm has better robustness, and the machine learning algorithm utilizes hand-crafted features with a higher lower line than DL. The ML classifier contains RF, MLP [47], XGBOOST, TCN, and VIT, with TCN showing the highest average accuracy, followed by CNN. Figure 9 and Figure 10 show the variation in performance of the nine baseline algorithms and the DFTrans algorithm for the eight transportation mode detection on the two datasets.

**Figure 9 sensors-22-08499-f009:**
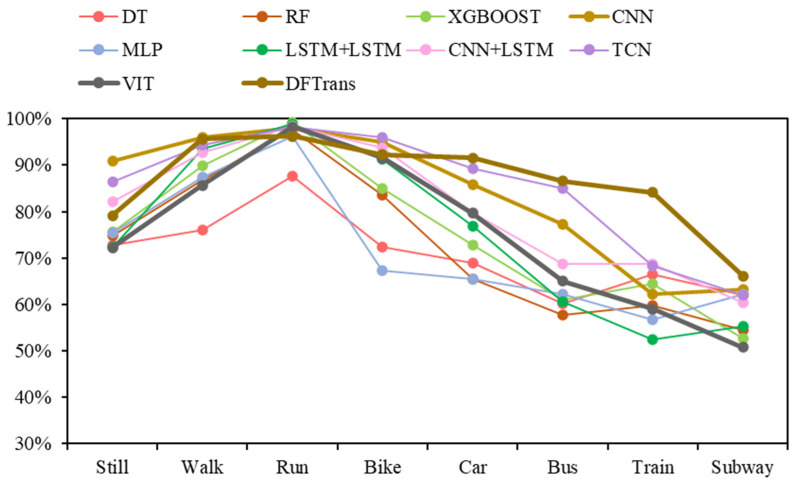
Variation in the performance of nine baselines and the DFTrans algorithm on eight transportation mode detection on the SHL dataset.

**Figure 10 sensors-22-08499-f010:**
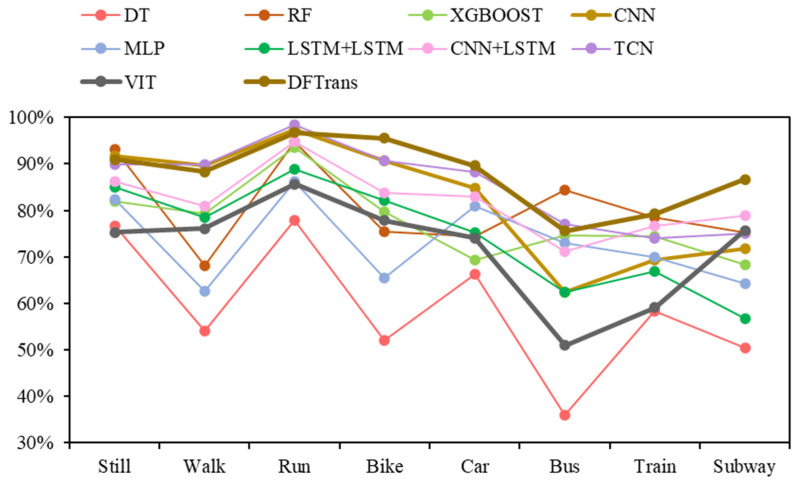
Variation in the performance of nine baselines and the DFTrans algorithm on eight transportation mode detection on the HTC dataset.

In addition, Figure 11 and Figure 12 show the confusion matrix of the experimental results on the two datasets, respectively. Figure 13 and Figure 14 show the recall, precision, and F1 scores of DFTrans on the two datasets, respectively.

### 4.7. Hyperparameter Fine-Tuning

We have fine-tuned the hyper-parameters several times to give a better and more objective performance of our DFTrans model. Temporal Block has a total of four convolutional layers. In our tuning process, we took the five best hyperparameter configurations a–e, and the performance of these five hyperparameter configurations is shown in Figure 15. F and K represent the filter and kernel sizes of the four convolutional layers on the Temporal Block in the DFTrans model, respectively. For Group a, F = [64, 64, 32, 32] and K = [3, 3, 3, 3]. For Group b, F = [64, 32, 64, 32] and K = [2, 2, 3, 3]. For Group c, F = [32, 32, 32, 32] and K = [2, 2, 2, 2]. For Group d, F = [128, 128, 32, 32] and K = [3, 3, 2, 2]. For Group e, F = [32, 32, 32, 32] and K = [3, 3, 3, 3].

As can be seen from Figure 15, the highest recognition performance was obtained for the “e” group parameter configuration. The train and metro modes both run smoothly on rails and are therefore the most challenging of the eight modes of transport to identify.

### 4.8. Impact of Each Sensor on Model Training Results

We included experiments with different sensor variables to verify the scalability of the DFTrans model. As shown in Figure 16, LA is short for linear accelerometer, LAG is short for linear accelerometer + gyroscope, LAGM is short for linear accelerometer + gyroscope + magnetometer, and LAGMP is short for linear accelerometer + gyroscope + magnetometer + barometric pressure. On the SHL dataset, the average accuracy of LA, LAG, LAGM, and LAGMP was 77.10%, 77.68%, 81.38%, and 86.01%, respectively. By adding the sensor, the accuracy of the model was improved from about 0.58% to 4.63%. As shown in Figure 17, the average transport mode detection accuracy of LA, LAG, LAGM, and LAGMP was 76.60%, 79.57%, 81.49%, and 87.86%, respectively, for the HTC dataset. By adding the sensor, the accuracy of the model was improved by about 1.92% to 6.37%. Through experiments with sensor variables, we have found that DFTrans can learn more about its functionality with more sensor variables and that barometric sensor data has a significant impact on improving DFTrans recognition.

### 4.9. Calculation Complexity

The detailed configuration of our experimental equipment has been described in Table 3. Table 8 shows the time complexity and space complexity of the different algorithms. In the experimental tests, all baseline algorithms and DFTrans models were trained with epochs set to 140.

As shown in Table 8, although the time complexity of DFTrans is higher than that of decision tree and random forest, it also improves the performance considerably because DFTrans takes a significant amount of time by automatically learning the data features. Although DFTrans has a higher spatial complexity than convolutional neural network, CNN + LSTM and VIT, the time complexity of DFTrans is much smaller than theirs. The forward transfer process of input information for all time steps of DFTrans is conducted in parallel and is therefore faster than LSTM. RNN forward transmission needs to wait for the completion of forward transmission at an earlier time step. The DFTrans algorithm saves about 3 times more time than the VIT algorithm because the Temporal Block shares the unstable high-frequency components, and the Attention Block only handles the stable low-frequency components.

## 5. Conclusions

In this paper, we present DFTrans, a novel deep-learning model for transportation mode detection. DFTrans uses the discrete wavelet transform to obtain the low- and high-frequency components of traffic sequences to accurately capture the temporal and spatial correlation between the low- and high-frequency components of long- and short-term patterns, by automatically learning the features of sensors integrated into a smartphone for transportation mode detection. DFTrans achieved an average accuracy of 86.69% on the SHL dataset and 87.51% on the HTC dataset. This is an improvement of 1.73% and 2.08% over the average accuracy of the best baseline on the two datasets, respectively. With different channels learning different sensor features in the Temporal Block and Attention Block, DFTrans can support a wider range of heterogeneous sensors. In addition, Temporal Block allows for massively parallel processing and requires less memory to train long sequences. As a result, DFTrans has greater utility in transportation mode detection. In the future, we will develop a cloud-based client/server framework to provide robust, lightweight, and accurate data. We will further increase the accuracy and responsiveness of the DFTrans model and evaluate the generalization capabilities across a wider range of environments.

## Figures and Tables

**Figure 1 sensors-22-08499-f001:**
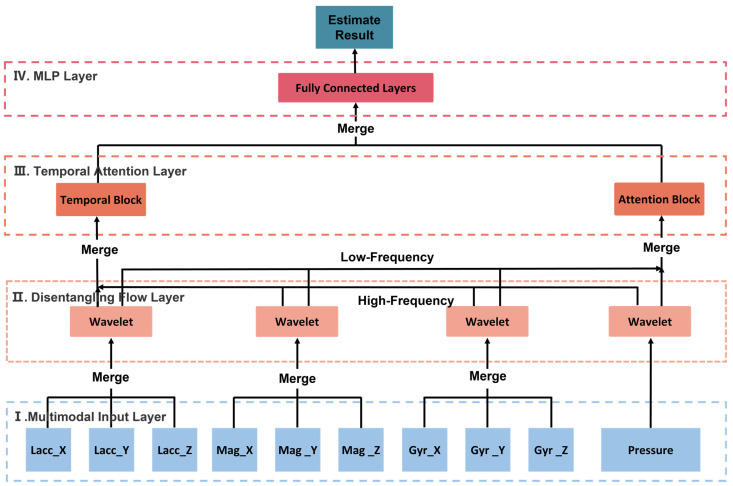
Overall overview diagram of our proposed DFTrans model. The pre-processed sensor acquisition data enters the DFTrans from the Multimodal Input Layer and enters the Disentangling Flow Layer for the separation of high-frequency and low-frequency components, respectively. The high-frequency components are processed in the Temporal Block, and the low-frequency components are processed in the Attention Block. Finally, it is processed by the MLP layer, and the classification results are output.

**Figure 2 sensors-22-08499-f002:**
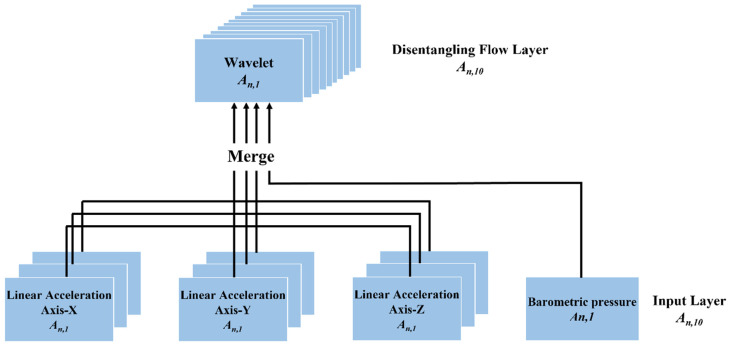
Multimodal Input Layer structure. The three inertial sensors’ x, y, and *z*-axis elements are fused and fed into the Disentangling Flow Layer. The barometric sensor tensor is fed directly into the Disentangling Flow Layer.

**Figure 3 sensors-22-08499-f003:**
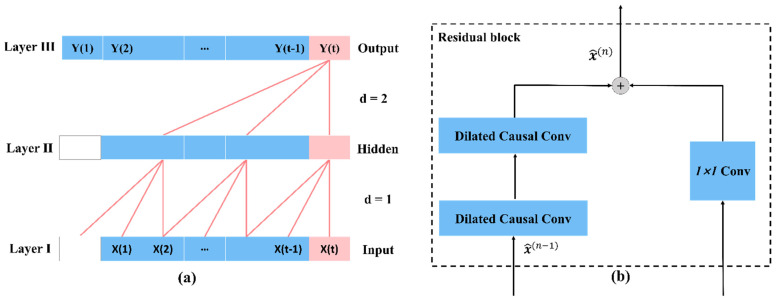
Structural elements in DFTrans. (**a**) Dilated causal convolution where the perceptual field can cover the input sequence. (**b**) To keep the size of the input and output the same, add a 1 × 1 convolution.

**Figure 4 sensors-22-08499-f004:**
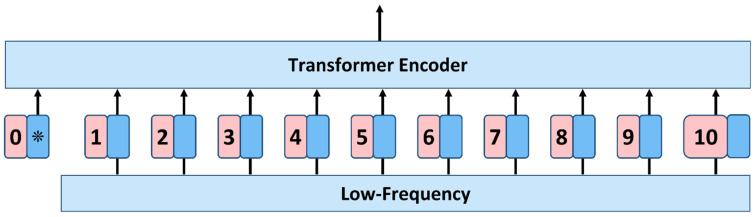
Structure of Attention Block. We add a positional embedding to the low-frequency component of each sensor and feed the resulting vector sequence into a Transformer encoder. We add learnable ‘classification marker’ to the sequence: * Extra learnable [class] embedding.

**Figure 5 sensors-22-08499-f005:**
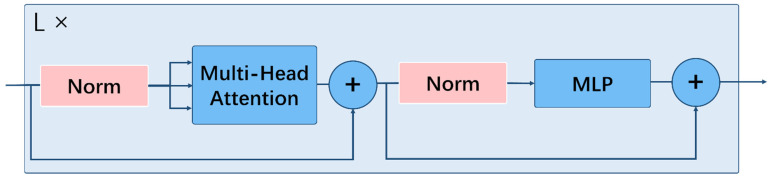
Structure of the Transformer Encoder in the attention block.

**Figure 6 sensors-22-08499-f006:**
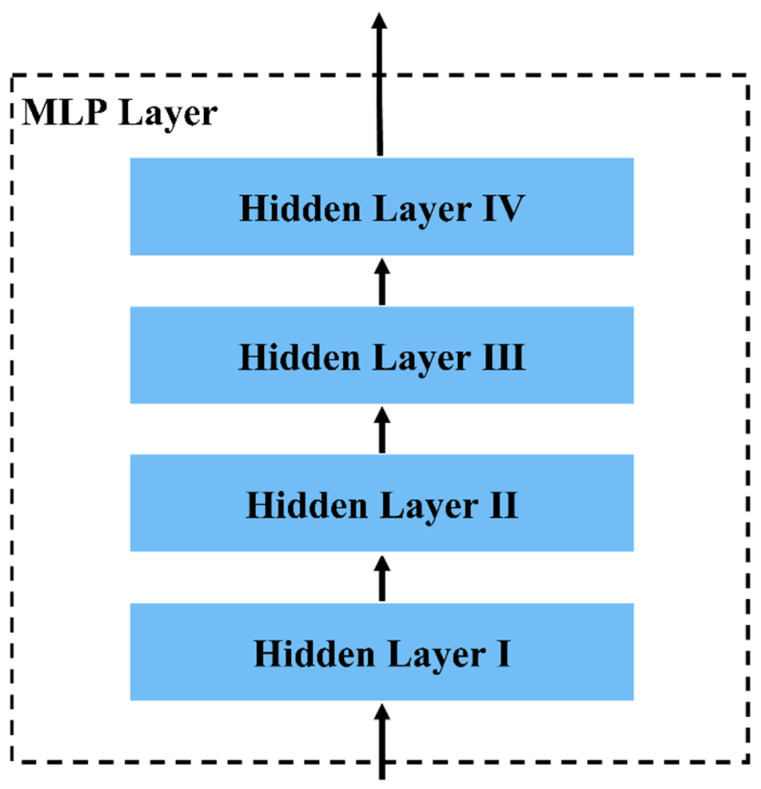
The structure of the MLP layer. The final output of the MLP layer is eight classifications.

**Figure 7 sensors-22-08499-f007:**
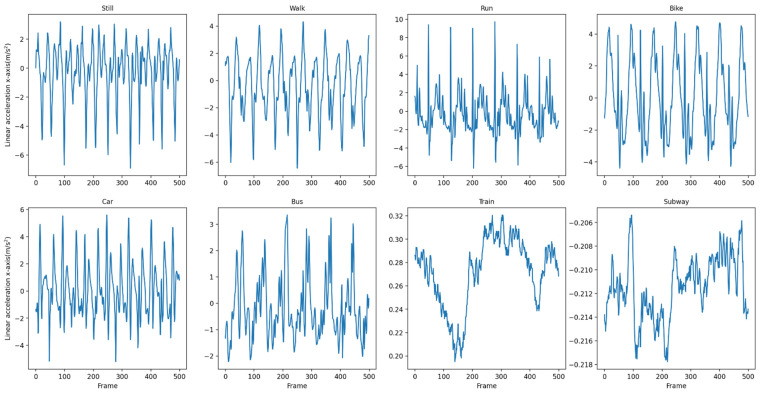
*X*-axis data for linear acceleration in the SHL dataset.

**Figure 8 sensors-22-08499-f008:**
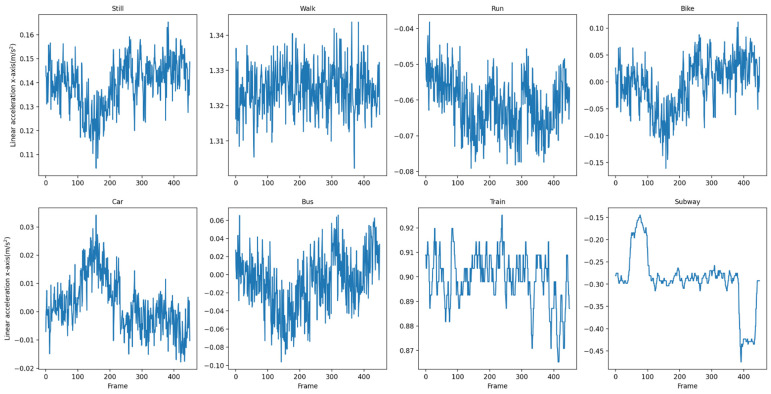
*X*-axis data for linear acceleration in the HTC dataset.

**Figure 11 sensors-22-08499-f011:**
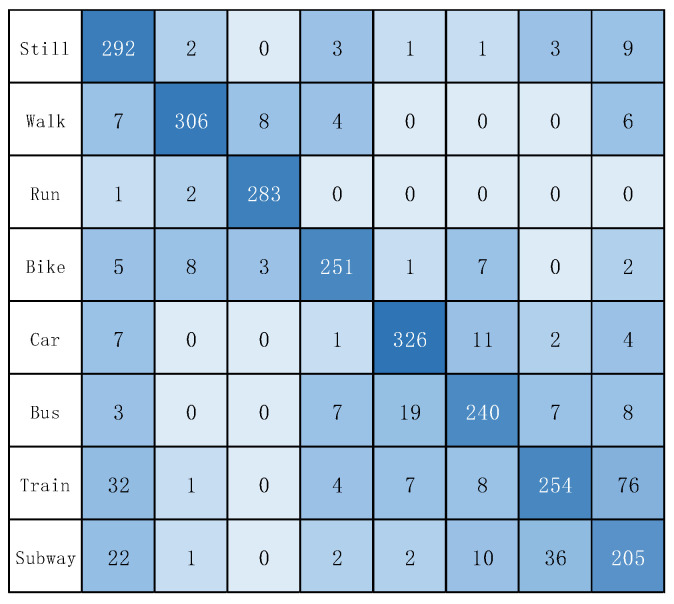
The confusion matrix of DFTrans on the SHL dataset.

**Figure 12 sensors-22-08499-f012:**
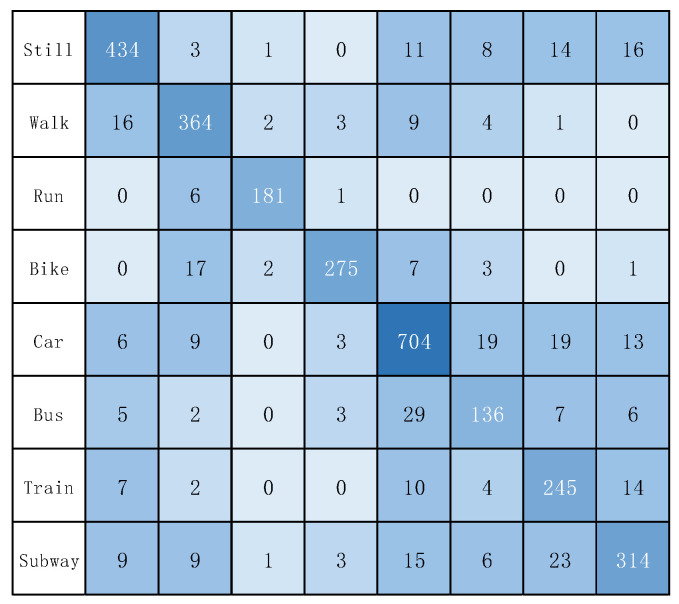
The confusion matrix of DFTrans on the HTC dataset.

**Figure 13 sensors-22-08499-f013:**
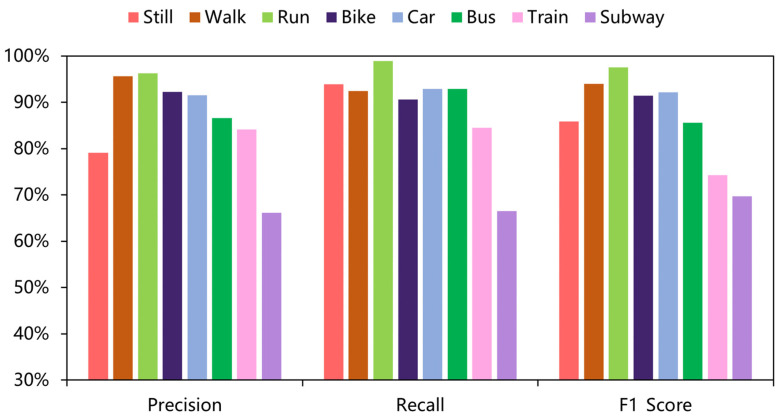
Performance evaluation results of the DFTrans model on the SHL dataset for eight classifications.

**Figure 14 sensors-22-08499-f014:**
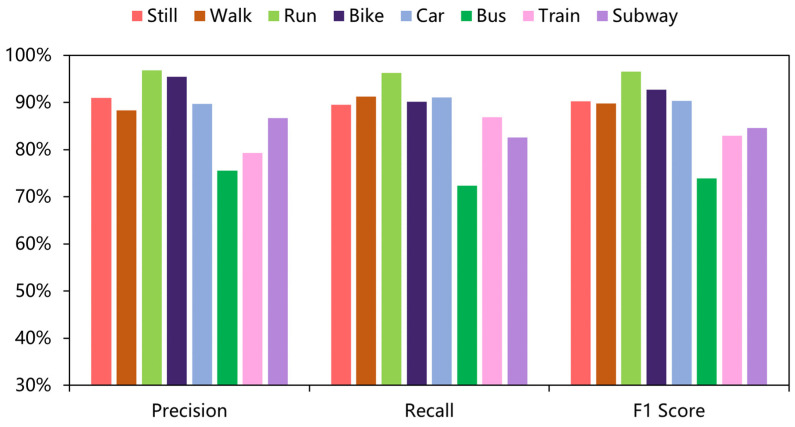
Performance evaluation results of the DFTrans model on the HTC dataset for eight classifications.

**Figure 15 sensors-22-08499-f015:**
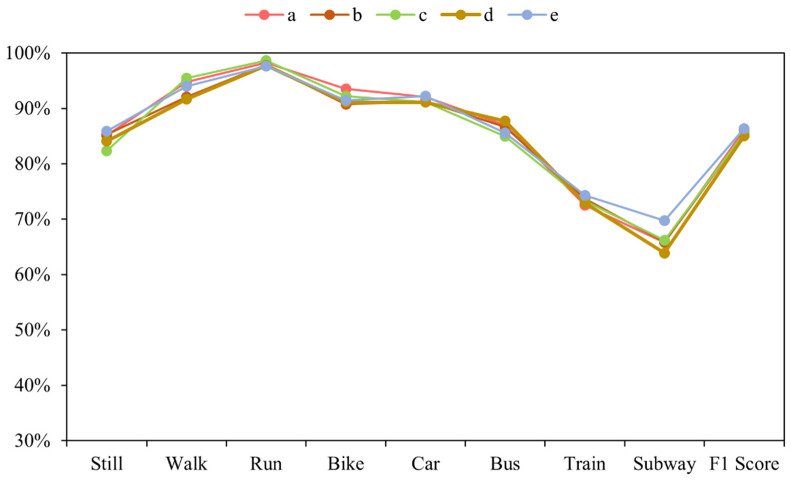
F1 scores were obtained by setting four different sets of hyperparameters (a–e) in the Temporal Block.

**Figure 16 sensors-22-08499-f016:**
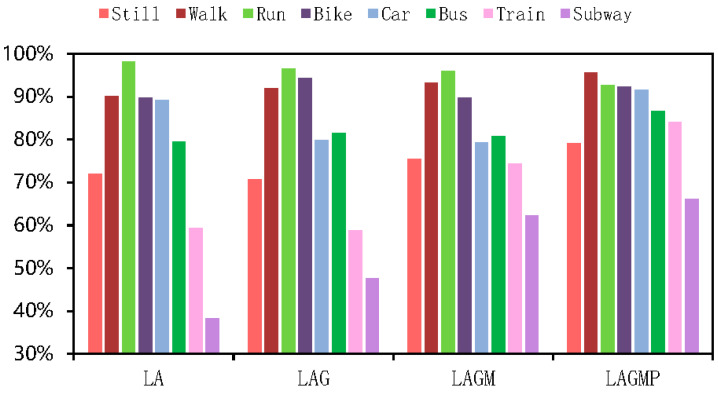
Precisions of DFTrans obtained using different sensor variables on the SHL dataset.

**Figure 17 sensors-22-08499-f017:**
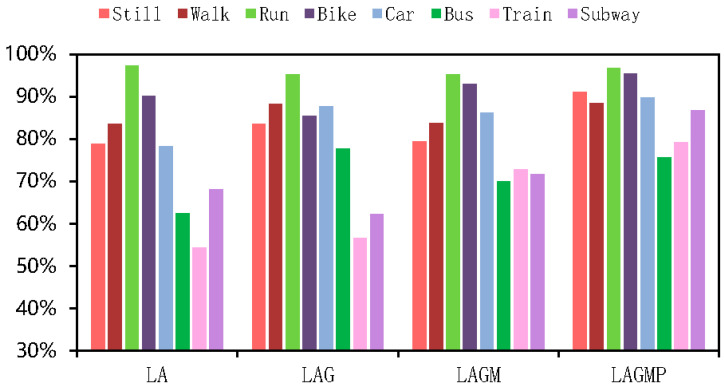
Precisions of DFTrans obtained using different sensor variables on the HTC dataset.

**Table 1 sensors-22-08499-t001:** Advantages and disadvantages of related work.

Paper	Data Sources	Strengths	Weaknesses
Inferring transportation modes from GPS trajectories using a convolutional neural network [21]	GPS	The layout of the CNN input layer was designed to improve the quality of the GPS logs through several data pre-processing steps to achieve an optimal CNN architecture with a maximum accuracy of 84.8%.	The process is more complex, uses a single dataset and has less application value.
Assessing the role of geographic context in transportation mode detection from GPS data [6]	GPS	Bridging the gap in understanding geography in transportation mode detection, the addition of contextual information specific to urban geography can improve the predictive accuracy of traffic pattern detection models.	A generalized model combining GPS data from multiple cities may still be useful for predicting patterns from trip data where knowledge of the geography of the study area is limited.
Transportation mode identification with GPS trajectory data and GIS information [22]	GPS, GIS	Using GIS information to improve inference accuracy while ensuring that the algorithm is easy to use on mobile devices.	Little improvement in accuracy, but increased consumption of data storage and computing resources.
Transportation mode detection using mobile phones and GIS information [23]	GIS	The study can better distinguish between these three difficult modes of transport: bus, car, and train.	The machine learning used is not ideal for other modes of transportation mode detection.
Smartphone transportation mode recognition using a hierarchical machine learning classifier and pooled features from time and frequency domains [2,9]	GPS	Transforming the time domain features to the frequency domain also adds new features in a new space and provides more control on the loss of information.	Data are excluded from this article because system use may drain the smartphone’s battery and signal may be lost in certain areas.
Combining Residual and LSTM Recurrent Networks for Transportation Mode Detection Using Multimodal Sensors Integrated in Smartphones [28]	IMU	Residual units are introduced to speed up learning, and attention models are used to learn the importance of different features and different time steps to improve recognition accuracy.	Increased consumption of data storage and computing resources and longer computation times.
Benchmarking the SHL Recognition Challenge with Classical and Deep-Learning Pipelines [29]	IMU	The study uses a more refined 8 modes of transport, with a more detailed division of the called modes of transport.	Only the performance of a single classical machine learning algorithm for transportation mode detection was compared.
End-to-end trajectory transportation mode classification using Bi-LSTM recurrent neural network [17]	IMU	The proposed classification process does not require any feature extraction process, but automatically learns features from the trajectories and uses them for classification.	Only the AUC aspect is described as superior and the overall performance is not fully evaluated.
CNN for human activity recognition on small datasets of acceleration and gyro sensors using transfer learning [30]	IMU	The difficulty was overcome: there was not enough training data for the target sensor marked as Hand.	The maximum recognition rate of the model is only about 82%, with no significant improvement.
Toward Transportation Mode Recognition Using Deep Convolutional and Long Short-Term Memory Recurrent Neural Networks [31]	IMU	The use of CNNs to learn appropriate and robust feature representations overcomes the reliance of machine learning on manual features.	The complexity of the algorithm is too high.
Calculating Travel Time across Different Travel Modes Using Bluetooth and WiFi Sensing Data [32]	WiFi	The technology detects personal electronic devices to determine people’s movements, rather than the traditional method of detecting a single pattern.	The accuracy is low, which is more limited for some applications in traffic engineering.
WiFi CSI Based Passive Human Activity Recognition Using Attention Based BLSTM [33]	WiFi	The attention mechanism is used to assign different weights to all learned features, enabling good performance to be achieved.	Too small a data set, weakly persuasive.
WiFi CSI based Human Activity Recognition Using Deep Recurrent Neural Network [34]	WiFi	A more robust relationship can be established between transportation mode detection and WiFi CSI than most existing WiFi CSI-based methods.	The process requires the extraction of two representative features from different statistical curves, with weak generalizability.

**Table 2 sensors-22-08499-t002:** Parameters specifically modified in baselines.

Algorithm	Parameters
DT	min_leaf_size = 800
RF	minleafsize = 800
XGBOOST	n_estimators = 800, max_depth = 7
CNN	C(32)-C(32)-C(64)
MLP	FC(64)-FC(128)-FC(256)-FC(512)-Softmax
Bi-LSTM	LSTM(128)-LSTM(64)-FC-Softmax
CNN + LSTM	C(64)-C(64)-C(64)-LSTM(128)-DNN(128) -Softmax
TCN	C(32)-C(32)-C(32)-C(32)-C(32) -C(32)
VIT	NumPatches = 251, ProjectionDim = 10, NumHeads = 3, TransformerLayers = 4

Note: C represents a convolutional 1D layer; FC represents a fully connected layer.

**Table 3 sensors-22-08499-t003:** Configuration for training the DFTrans model.

Name	Detail
CPU	Intel(R) Xeon(R) CPU @ 2.30 GHz
GPU	Tesla P100-PCIE-16GB
Memory	16GB
Framework	Keras
Operating System	Ubuntu 18.04.5 LTS
Python Environment	3.7.13

**Table 4 sensors-22-08499-t004:** Precisions of different algorithms on the SHL dataset.

	DT	RF	XGBOOST	MLP	CNN	CNN + LSTM	Bi-LSTM	TCN	VIT	DFTrans
Still	72.87%	74.75%	75.60%	75.45%	90.97%	82.14%	66.75%	90.65%	71.72%	79.13%
Walk	76.01%	86.77%	89.98%	87.43%	95.97%	92.82%	89.33%	96.34%	84.34%	95.63%
Run	87.58%	97.68%	99.34%	96.30%	98.02%	98.50%	98.01%	98.86%	98.00%	96.26%
Bike	72.32%	83.56%	85.10%	67.37%	95.05%	93.81%	85.67%	91.62%	90.57%	92.28%
Car	68.93%	65.42%	72.76%	65.49%	85.81%	79.67%	70.16%	93.66%	77.13%	91.57%
Bus	60.30%	57.73%	61.11%	62.17%	77.27%	68.83%	58.02%	81.82%	68.72%	86.64%
Train	66.57%	59.78%	64.46%	56.64%	62.21%	68.78%	50.06%	68.81%	58.98%	84.11%
Subway	61.97%	54.51%	52.63%	62.24%	63.16%	60.49%	46.42%	66.77%	52.16%	66.13%

**Table 5 sensors-22-08499-t005:** Precisions of different algorithms on the HTC dataset.

	DT	RF	XGBOOST	MLP	CNN	CNN + LSTM	Bi-LSTM	TCN	VIT	DFTrans
Still	76.74%	93.22%	81.91%	82.47%	91.76%	86.31%	69.71%	92.49%	75.32%	90.99%
Walk	54.12%	68.04%	79.41%	62.54%	89.66%	81.03%	89.92%	87.48%	76.07%	88.35%
Run	77.98%	94.83%	93.62%	86.22%	97.34%	94.76%	97.05%	96.40%	88.85%	96.79%
Bike	52.06%	75.42%	79.74%	65.54%	90.79%	83.89%	83.89%	91.91%	77.91%	95.49%
Car	66.23%	74.50%	69.36%	80.94%	84.85%	82.99%	70.16%	87.45%	78.33%	89.68%
Bus	36.07%	84.48%	74.71%	73.01%	62.50%	71.22%	63.02%	82.93%	54.97%	75.56%
Train	58.38%	78.53%	74.50%	69.88%	69.44%	76.60%	51.27%	74.60%	61.81%	79.29%
Subway	50.42%	75.22%	68.36%	64.35%	71.88%	78.99%	51.55%	71.59%	77.74%	86.74%

**Table 6 sensors-22-08499-t006:** Accuracy of different algorithms on the SHL dataset.

	RF	XGBOOST	CNN	MLP	LR + MLP	Bi-LSTM	CNN + LSTM	TCN	VIT	DFTrans
Still	75.95%	85.40%	81.46%	84.74%	86.18%	72.45%	80.16%	86.41%	72.30%	93.89%
Walk	89.96%	84.65%	96.34%	60.15%	79.70%	93.62%	94.76%	94.33%	85.71%	92.45%
Run	89.56%	89.43%	98.29%	87.77%	85.64%	98.91%	98.29%	98.30%	98.27%	98.95%
Bike	83.63%	64.90%	94.31%	57.38%	80.33%	91.40%	88.92%	95.94%	91.56%	90.61%
Car	58.18%	90.09%	90.38%	83.31%	76.33%	76.82%	81.92%	89.30%	79.63%	92.88%
Bus	60.59%	38.28%	76.83%	59.57%	48.93%	60.67%	69.21%	84.97%	65.02%	84.51%
Train	58.60%	69.95%	66.59%	80.85%	60.99%	52.49%	54.20%	68.39%	59.04%	66.49%
Subway	63.37%	68.83%	65.26%	46.84%	65.53%	55.34%	57.10%	62.06%	50.76%	73.74%

**Table 7 sensors-22-08499-t007:** Accuracy of different algorithms on the HTC dataset.

	RF	XGBOOST	CNN	MLP	LR + MLP	Bi-LSTM	CNN + LSTM	TCN	VIT	DFTrans
Still	93.90%	89.14%	89.69%	87.22%	84.33%	84.94%	89.07%	89.98%	75.32%	89.48%
Walk	68.27%	75.44%	91.23%	67.42%	75.69%	78.47%	86.72%	89.92%	76.07%	91.23%
Run	94.74%	92.05%	97.87%	92.55%	92.55%	88.88%	94.15%	98.40%	85.71%	96.28%
Bike	73.02%	69.77%	89.84%	67.21%	82.95%	82.13%	87.87%	90.71%	77.85%	90.16%
Car	75.04%	81.56%	88.23%	81.50%	78.00%	75.31%	83.83%	88.22%	74.08%	91.07%
Bus	92.59%	52.68%	70.74%	62.23%	42.02%	62.50%	58.51%	77.18%	50.97%	72.34%
Train	78.88%	73.99%	76.95%	77.31%	47.16%	67.01%	75.18%	73.93%	59.11%	86.88%
Subway	73.50%	71.88%	78.42%	61.05%	67.37%	56.78%	71.05%	75.12%	75.74%	82.63%

**Table 8 sensors-22-08499-t008:** The training time and parameter size of baselines.

Algorithm	Platform Type	Training Time	Parameter Size
DT	GPU	75 s	-
RF	GPU	90.3 s	-
XGBOOST	GPU	6653.5 s	-
MLP	GPU	140.4 s	1,338,248
CNN	GPU	1200.1 s	70,348
CNN + LSTM	GPU	64,080 s	125,568
TCN	GPU	447.3 s	76,688
VIT	GPU	3204.4 s	53,948
DFTRANS	GPU	786.1 s	227,602

## Data Availability

The SHL dataset [48] for this article is publicly available on the website http://www.shl-dataset.org/dataset (accessed on 1 May 2022) and the HTC dataset for this article is required to see reference [8].

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
