# Peer review of "DFTrans: Dual Frequency Temporal Attention Mechanism-Based Transportation Mode Detection"

_sensors, 2022, doi:10.3390/s22218499_

Round 1

Reviewer 1 Report

1.     This is an interesting study and the authors have collected a unique dataset. The paper is generally well written and structured.

2.     More generally, I suggest to focus the manuscript on the scientific results rather than on the innovation in engineering.

3.     Abstract should be description of the problem. Seems it is not much clear.

4.     Related works is summarized only but the authors did not criticize them. Recommendations to use these methods are not exist. The problems of related work are not clear. In other words, why did the authors propose new method? Or why related works are not used in solving this problem. I think the last paragraph of related word might present related work problems then introduce why they propose new method.

5.     It is unclear how the proposed work addressed a research gap in the given domain Strong arguments are required.

6.     7. All the equations need to be described properly.

7.     What is the source data details not found anywhere.

8.     Results achieved. It describes clearly what has been done before on the
problem, and what is new.

9.     The main idea that how the authors improved the base-line method is not obvious

Author Response

We have uploaded our response letter as a PDF file, please refer to it. Thank you.

Reviewer 2 Report

This empirically  investigation study seems interesting    but there are some serious concerns that authors need to refine the revised version Major revision,

1. The novelty of this study is missing, authors need to highlight the novel contribution in detail.

2.In introduction section, authors need to highlight the representation of IMU data

3.In introduction section, the contribution part clearly missed the novel contribution of this study. Furthermore, in third contribution evaluation of DFTrans is missed.

4.In Section 2 Related work, authors need to add the tabular form of SWOT analysis, that clearly represent the gap in the previous techniques.

5. Authors clearly highlight the signifinace of this study in real time problems and must embed the future directions of this study  

Author Response

(The authors gave the same response as above.)

Reviewer 3 Report

The paper is generally well-written. On the other hand, the paper should be revised by considering the following issues.

MAJOR ISSUES:

1) The related work of the paper should be improved by adding more papers which applies machine learning techniques on similar problems like transmitter detection via RF Fingerprinting. Some other physical layer security methods like RF fingerprinting techniques can perform very well for transmitter detection/identification problem in the cyberphysical/transportation systems. What is the motivation of the proposed approach in this paper? For this purpose, I strongly recommend the authors should include the following three papers in their related work in order to clarify not only the main contribution but also motivation of this paper in the related literature.

O. M. Gul, M. Kulhandijan, B. Kantarci, A. Touazi, C. Ellement, C. D’Amours, "Fine-Grained Augmentation Approach for RF Fingerprinting under Impaired Channels", IEEE International Workshop on Computer Aided Modeling and Design of Communication Links and Networks 2022 (IEEE CAMAD 2022), 2-4 November 2022, France, pp. 1-6.

G. Reus-Muns, D. Jaisinghani, K. Sankhe and K. R. Chowdhury, "Trust in 5G Open RANs through Machine Learning: RF Fingerprinting on the POWDER PAWR Platform," GLOBECOM 2020 - 2020 IEEE Global Communications Conference, 2020, pp. 1-6.

+The outline of the paper should be given more clearly at the end of introduction section.

+The algorithm should be also given as pseudocode format.

+Size of Figures and Tables should be improved such that they will not exceed the margins.

MINOR ISSUES

+Typos and grammatical errors should be fixed.

Author Response

(The authors gave the same response as above.)

Round 2

Reviewer 3 Report

The authors addressed all my concerns. It can be accepted for publication in its present form.